# Prognostic Value of Holter Monitoring in Light Chain Amyloidosis

**DOI:** 10.3390/jcm12237457

**Published:** 2023-12-01

**Authors:** Yutong Sun, Qinghao Zhao, Yang Liu, Lei Wen, Xuelin Dou, Jin Lu, Jian Liu

**Affiliations:** 1Department of Cardiology, Peking University People’s Hospital, Beijing 100044, China; sunyt@bjmu.edu.cn (Y.S.); qhzhao@hsc.pku.edu.cn (Q.Z.); 2Center for Cardiovascular Translational Research, Peking University People’s Hospital, Beijing 100044, China; 3Beijing Key Laboratory of Early Prediction and Intervention of Acute Myocardial Infarction, Peking University People’s Hospital, Beijing 100044, China; 4Department of Hematology, Peking University People’s Hospital, Beijing 100044, China; pkupuliuyang@vip.sina.com (Y.L.); cillits@163.com (L.W.); dxldw@163.com (X.D.); 5Beijing Key Laboratory of Hematopoietic Stem Cell Transplantation, Peking University, Beijing 100044, China

**Keywords:** atrial tachycardia, conduction delay, Holter monitoring, prognosis, systemic light chain amyloidosis

## Abstract

(1) Background: To evaluate the predictive value of Holter monitoring for overall survival (OS) of patients with light chain amyloidosis (AL amyloidosis). (2) Methods: 137 patients with newly diagnosed AL amyloidosis who underwent Holter monitoring within 6 months of diagnosis were included. The primary outcome was OS. Landmark analysis was conducted at one-year follow-up. Independent predictors were determined using the log-rank test and multivariate Cox regression analysis. (3) Results: 131 (95.6%) patients received non-transplant therapy, and 32 (23.4%) underwent daratumumab-based chemotherapy. After a median follow-up of 20.3 months, 47 deaths occurred. Atrial tachycardia (AT), conduction delay, and non-sustained ventricular tachycardia (NSVT) were associated with poor OS one year beyond diagnosis in univariate analyses (patients with vs. without AT: 57.3% [95% confidence interval (CI): 47.2–67.4] vs. 81.0% (95% CI: 74.8–87.2), *p* = 0.039; patients with vs. without NSVT: 33.3% (95% CI: 8.5–58.1) vs. 75.3% (95% CI: 69.8–80.8), *p* = 0.024; patients with vs. without conduction delay: 41.7% (95% CI: 24.4–59.0) vs. 75.4% (95% CI: 69.7–81.1), *p* = 0.003]. AT [hazard ratio (HR): 2.6; 95% CI: 1.0–6.5; *p* = 0.049) and conduction delay (HR: 4.3; 95% CI: 1.3–14.3; *p* = 0.016) were independent predictors of OS after accounting for age and 2012 Mayo stage. (4) Conclusion: AT and conduction delay in Holter monitoring are independent predictors of poor OS one year beyond diagnosis in AL amyloidosis.

## 1. Introduction

AL amyloidosis, also known as primary systemic light chain amyloidosis, is characterized by amyloid deposition composed of monoclonal immunoglobulin light chains, leading to progressive multi-organ dysfunction [1,2]. The clinical manifestations vary depending on the affected organs, encompassing restrictive cardiomyopathy, nephrotic syndrome, hepatic failure, and peripheral/autonomic neuropathy. According to the National Comprehensive Cancer Network (NCCN) guidelines of AL amyloidosis, key elements in the diagnostic process include the identification of clinical features consistent with amyloidosis, detection of monoclonal protein by serum and urine electrophoresis, confirmation of monoclonal light chains through immunofixation, and tissue biopsy showing amyloid deposits [3]. The Mayo 2012 staging system is a commonly used staging system for AL amyloidosis, comprising three indicators: N-terminal pro-brain natriuretic peptide (NT-pro BNP), serum troponin T/I, and serum immunoglobulin free light chain difference (dFLC). This staging system categorizes patients into four stages with significant differences in overall survival (OS), with the poorest stage (i.e., stage IV) having a median OS of only 6 months [4]. 

Cardiac involvement is observed in 50% to 70% of patients with AL amyloidosis [5,6,7] and is the most important indicator of poor prognosis, including heart failure, reduced quality of life, and death [8]. Patients with cardiac amyloidosis (CA) are also at higher risk of various arrhythmias, with multifactorial pathogenesis including amyloid deposition and cytotoxicity. Common arrhythmias observed in AL amyloidosis patients include atrial fibrillation (AF), atrial tachycardia, ventricular arrhythmias, and conduction delay, which are often highly symptomatic, poorly tolerated, and require closer monitoring [9,10,11,12,13,14,15].

Twenty-four-hour Holter monitoring can detect transient and dynamic arrhythmias with a higher sensitivity than a standard 10 s electrocardiogram (ECG). However, its predictive value of OS in AL amyloidosis is unclear. Palladini et al. reported that ventricular couplets (VC) on 24 h Holter monitoring may have prognostic significance of OS in a study of 51 patients with AL amyloidosis [16]. Sidana et al. assessed 239 patients with AL amyloidosis who mainly received autologous hematopoietic stem cell transplantation (ASCT) and found that atrial fibrillation (AF) and non-sustained ventricular tachycardia (NSVT) were associated with poorer short-term OS [17]. 

The treatment landscape of AL amyloidosis has changed dramatically in recent years. The traditional treatment for AL amyloidosis has involved therapies typically used for multiple myeloma, targeting plasma cells. The most common regimen is a combination of bortezomib, cyclophosphamide, and dexamethasone. However, despite advancements in previous standard therapies, the prognosis for this disease, particularly in terms of OS, remains suboptimal [6,18,19]. With the introduction of the anti-CD38 monoclonal antibody (daratumumab)-based combination regimen in the front-line setting, significant improvements in hematologic complete response (CR), organ responses [20,21], and potentially enhanced OS have been realized for AL amyloidosis patients. 

In this context, it is unknown whether 24 h Holter monitoring continues to forecast OS in AL amyloidosis. Therefore, we undertook this study to evaluate the prognostic value of continuous dynamic Holter monitoring for OS in patients with AL amyloidosis.

## 2. Materials and Methods

### 2.1. Study Participants

Peking University People’s Hospital is the largest center experienced in hematopoietic stem cell transplantation (HSCT) in Asia. We consecutively enrolled all the patients newly diagnosed with AL amyloidosis at this center between 28 December 2011 and 23 November 2022; patients also underwent 24 h Holter monitoring. The diagnosis of AL amyloidosis was established through the identification of Congo-red-positive fibril deposition in the biopsy. Then, the AL subtype was determined via immunohistochemistry, immunofluorescence, and immunoelectron microscopy [22,23]. In addition, the updated International Myeloma Working Group (IMWG) consensus criteria were used to exclude the presence of active multiple myeloma [24]. Specifically, if bone marrow plasma cells were more than 10% and less than 60%, and computerized tomography or whole-body magnetic resonance imaging indicated no evidence of bone destruction, patients were diagnosed with AL amyloidosis. If the interval between diagnosis and Holter monitoring was over 6 months or patients declined further follow-up, the patient was excluded. If a patient underwent Holter monitoring more than once during the 6-month inclusion window, the results other than the first Holter study were excluded. We calculated the sample size using R (Version 4.1.2, R Development Core Team, Vienna, Austria), with detailed methods described in the Appendix A. Institutional Review Boards at Peking University People’s Hospital approved the study protocol. Informed consent was obtained from all participants before study participation.

### 2.2. Data Collection and Definitions

Baseline demographics, examination findings, diagnosis, and treatment-related information were obtained from electronic medical records. All laboratory tests were conducted at the time of diagnosis. The 2012 Mayo Clinic prognostic staging system for AL amyloidosis was used to stratify patients by risk [4]. Cardiac involvement was defined as positive findings on endomyocardial biopsy, mean wall thickness >12 mm on echocardiography without other cardiac cause, or an elevated NT-pro BNP (>332 ng/L) in the absence of renal failure or AF [3]. Cardiac conduction delay included sinus node dysfunction, atrioventricular block (AV block), and conduction tissue disease [25]. Atrial tachycardia (AT) included focal and multifocal AT, but sinus tachycardia, macro-re-entrant atrial tachycardia, and AF were excluded [26]. NSVT was defined as three or more consecutive ventricular beats at a rate of >100 beats per minute (bpm), lasting <30 s. The presence of arrhythmia was defined by the occurrence of ≥1 episode during Holter monitoring. 

### 2.3. Primary Outcome

The primary outcome was OS. We obtained outcome data from patient visits, medical records, and telephone interviews at 30 days, six months, and annually following diagnosis.

### 2.4. Statistics Analysis

Continuous data were summarized as mean ± standard deviation (SD) or median and interquartile range (IQR) and compared using the Student’s *t*-test or Mann–Whitney U-test, as appropriate. Categorical data were summarized as counts and percentages and compared using the chi-square test. Given that our study included 137 patients, we used the Shapiro–Wilk test to examine normality.

Survival analysis was conducted using the Kaplan–Meier method, and survival curves were compared using the log-rank test. OS was calculated from the time of diagnosis until death or last follow-up. Patients who were still alive at the last follow-up were censored. To gain further insight into the prognostic impact of Holter findings on early and late survival, we compared OS at fixed time points of 1 and 5 years, respectively. We also performed a landmark survival analysis with a landmark point set at one year, where the entire follow-up period was divided into two parts (first and subsequent years), and survival analyses were conducted separately [27]. Factors found to be statistically significant in univariate analysis were entered into a multivariable Cox analysis of OS, which provided estimates of hazard ratios (HRs) and 95% confidence intervals (CIs). To avoid overfitting and take into consideration limited events and clinical relevance [17], multivariable models were adjusted by age and Mayo 2012 stage (a well-validated prognostic staging system for AL amyloidosis) [4]. All tests were two-tailed, and statistical significance was set at *p* < 0.05. Statistical analyses were performed using SPSS (Version 25.0, IBM-SPSS, Chicago, IL, USA). 

## 3. Results

### 3.1. Demographic and Clinical Characteristics

The demographic and clinical characteristics and 24 h Holter findings of the study’s 137 participants are summarized in Table 1. A total of 102 (74.5%) participants were male, and their mean age was 61.9 years (mean ± SD: 51.7–72.1). In this cohort, fourteen (10.2%) patients had concomitant coronary artery disease, forty-six (33.6%) patients had concomitant hypertension, seventeen (12.4%) patients had comorbid diabetes, and one (0.7%) patient had a history of chronic kidney disease. A total of 110 (80.3%) patients had cardiac involvement and 101 (73.7%) patients had ≥2 organs involved. A total of 57 (41.6%) patients were characterized as having an advanced Mayo 2012 stage (III and IV). A total of 131 (95.6%) patients received non-transplant therapy as front-line treatment, among which 32 (23.4%) patients received daratumumab-based chemotherapy. 

### 3.2. Holter Findings and Anti-Arrhythmic Therapies

The median time between diagnosis and Holter examination was 5.9 days (IQR 3.0 to 24.4). The median duration of Holter monitoring was 23 h 19 min (IQR 17 h 50 min to 28 h). Most patients had normal sinus rhythm (116 patients, 84.7%), followed by sinus with conduction delay (17 patients, 12.4%). Sustained ventricular tachycardia (VT) or ventricular fibrillation (VF) requiring cardioversion did not occur in any of the study participants. In the whole cohort, twenty-three (16.8%) patients were treated with anti-arrhythmic drugs, while four (2.9%) patients underwent pacemaker implantation following Holter monitoring (Table 2).

In the subgroup analysis of patients with and without CA, the prevalence of AT was higher in patients with CA compared to those without (58 (52.7%) vs. 7 (25.9%), *p* = 0.012). However, the proportion of patients receiving anti-arrhythmic therapies did not significantly differ, regardless of the presence or absence of CA (23 (20.97%) vs. 3 (11.1%), *p* = 0.245) (Table 3).

### 3.3. Survival Analyses

During a median follow-up of 20.3 months (IQR 9.2–55.7 months), 47 deaths (34.3%) were observed. We found a significant difference in median OS between patients with and without CA (log-rank *p*-value = 0.045) (Appendix A). The Kaplan–Meier curves illustrating the outcomes of CA and non-CA patients are presented in Appendix A.

We report on arrhythmias found to be statistically significant in univariate analyses of the whole cohort or through previous studies below. OS in the presence of atrial couplets, AT, AF, VC, NSVT, and conduction delay are reported in Table 3. Kaplan–Meier curves of AT, NSVT, and conduction delay are found in Figure 1A. OS did not differ significantly between patients with and without Holter abnormalities (log-rank *p*-values for atrial couplets, AT, AF, VC, NSVT, and conduction delay were all >0.05).

In the landmark analysis, there were directly different results for OS within the first year compared with that beyond the first year for patients with and without AT, NSVT, and conduction delay (Figure 1B and Table 4). At one year, patients with and without AT, NSVT, or conduction delay had similar OS (patients with vs. without AT: 82.3% (95% CI: 77.4–87.2) vs. 81.5% (95% CI: 76.9–86.1), *p* = 0.897; patients with vs. without NSVT: 78.9% (95% CI: 69.5–88.3) vs. 82.6% (95% CI: 79.0–86.2), *p* = 0.471; patients with vs. without conduction delay: 83.3% (95% CI: 74.5–92.1) vs. 81.8% (95% CI: 78.2–85.4), *p* = 0.974). However, between 1 and 5 years, significantly poorer OS was observed for patients with AT, NSVT, or conduction delay (patients with vs. without AT: 57.3% (95% CI: 47.2–67.4) vs. 81.0% (95% CI: 74.8–87.2), *p* = 0.039; patients with vs. without NSVT: 33.3% (95% CI: 8.5–58.1) vs. 75.3% (95% CI: 69.8–80.8), *p* = 0.024; patients with vs. without conduction delay: 41.7% (95% CI: 24.4–59.0) vs. 75.4% (95% CI: 69.7–81.1), *p* = 0.003). 

As for patients with and without atrial couplets, AF, or VC, there was no statistically significant difference in OS within the first year or beyond the first year among patients (in year 0–1: patients with vs. without atrial couplets: 86.2% (95% CI: 82.1–90.3) vs. 76.5% (95% CI: 71.0–82.0), *p* = 0.156; patients with vs. without AF: 60.0% (95% CI: 38.1–81.9) vs. 82.7% (95% CI: 79.3–86.1), *p* = 0.184; patients with vs. without VC: 80.0% (95% CI: 72.7–87.3) vs. 82.8% (95% CI: 79.1–86.5), *p* = 0.434; in years 1–5: patients with vs. without atrial couplets: 70.0% (95% CI: 62.5–77.5) vs. 74.4% (95% CI: 66.3–82.5), *p* = 0.768; patients with vs. without AF: 100.0% (95% CI: 100.0–100.0) vs. 70.8% (95% CI: 65.1–76.5), *p* = 0.330; patients with vs. without VC: 77.5% (95% CI: 65.9–89.1) vs. 71.6% (95% CI: 65.6–77.6), *p* = 0.779)).

In multivariate analyses, after adjusting for age and 2012 Mayo stage, the presence of Holter abnormalities had no prognostic significance for OS during the first year (AT: HR: 1.0; 95% CI: 0.4–2.4; *p* = 0.991; NSVT: HR: 0.6; 95% CI: 0.2–2.3; *p* = 0.505; conduction delay: HR: 0.8; 95% CI: 0.2–2.9; *p* = 0.776) (Table 4). Moreover, AT and conduction delay continued to be significant predictors of OS in patients who were still alive beyond the first year (AT: HR: 2.6; 95% CI: 1.0–6.5; *p* = 0.049; conduction delay: HR: 4.3; 95% CI: 1.3–14.3; *p* = 0.016), while a significant association was no longer found for NSVT (HR: 2.0; 95% CI: 0.6–7.1; *p* = 0.275) (Table 5).

## 4. Discussion

To our knowledge, our study is the first to identify AT and conduction delay as independent risk factors for OS in AL amyloidosis and to demonstrate the prognostic value of 24 h Holter monitoring for OS beyond one year. Our study findings also suggest that these significant risk factors are not independently associated with OS in the first year of diagnosis. Moreover, as our study participants mainly underwent non-transplant therapy, of which over 20% received anti-CD38 chemotherapy, our results can be generalized to a larger population given the stringent indications of ASCT and the widespread use of anti-CD38 monoclonal antibodies.

### 4.1. The Prognostic Value of AT

Amyloid fibrils result from misfolding of overproduced immunoglobulin light chain deposits in the myocardium, promoting cardiac stiffness, hypertrophy, and diastolic dysfunction, elevating ventricular-filling pressures, and inducing atrial arrhythmias like AT and AF early in the course of the disease [8,12,28,29]. According to our study findings, AT occurs more frequently in patients with heart involvement than in those without (52.7% vs. 25.9%, *p* = 0.012). Thus, AT may be an early Holter biomarker of cardiac amyloidosis, foreshadowing a higher risk of myocardial hypertrophy, diastolic dysfunction, and poorer future outcomes.

Studies have reported that tissue infiltration by amyloid fibrils may trigger the coagulation cascade and cause intracardiac thrombi [30,31]. In a retrospective series of 116 autopsy or explanted cases of cardiac amyloidosis, 51% of patients with light chain cardiac amyloidosis (AL-CA) had intracardiac thrombi, and 26% had fatal embolic events [32]. Further, a multicenter retrospective study of 406 patients with CA reported that 7.6% of patients experienced systemic embolic events that manifested as stroke, transient ischemic attack, or extracranial embolic events [33]. These findings indicate that AT may be a marker of intracardiac thrombosis in patients with AL amyloidosis, leading to poor OS.

### 4.2. The Prognostic Value of Conduction Delay

Abnormal protein deposition in the myocardium also results in disarrangement of the myocardial architecture, disrupting the transmission of electrical impulses along conduction fibers [34]. This leads to conduction delay in ECG, mainly manifesting as AV block and bundle branch blocks [35,36]. An Italian study of 233 patients [37] found that AV block, bundle-branch block, and hemiblock were present in 18%, 23%, and 29% of patients with AL-CA, respectively. In a study of 58 patients with amyloidosis (20 with AL amyloidosis), patients with conduction delay had a higher incidence of pacemaker implantation at least one year after diagnosis [36]. Further, in a study of 405 patients with amyloidosis (119 with AL), PR interval > 200 ms and QRS interval > 120 ms predicted future pacemaker implantation [38]. Studies have also demonstrated that cardiac pacing primarily provides symptom relief without changing the survival of amyloidosis [39,40]. In summary, conduction delay predicts future pacemaker implantation and, thus, poorer lifelong outcomes (including OS) in patients with AL amyloidosis.

### 4.3. Contradictory Findings

Our results differ from Palladini et al.’s study of 51 patients with AL amyloidosis [16], which concluded that VC was an independent predictor of sudden death after adjusting for echocardiography findings. These contrary findings may be explained by differences in the covariates (e.g., Mayo stage) adjusted for in multivariate analyses and the study’s small sample size (which limited its power).

Our findings are also inconsistent with Sidana et al.’s study of 239 patients with AL amyloidosis, which found that AF, NSVT, and VC independently predicted short-term OS [17]. Possible explanations for these discrepancies in findings include the fact that Sidana et al.’s results were based on subgroup analyses of patients whose Holter monitoring was mainly driven by symptoms or arrhythmia history, while our findings were based on routine Holter examination at the time of diagnosis. Different timing may have also resulted in higher rates of arrhythmias in Sidana et al.’s study, especially in the case of ventricular arrhythmia and AF. Another explanation for the differing findings between our study and Sidana et al. is that 23.4% of the patients in our cohort received daratumumab-based therapy, whereas, in their non-transplant subgroup, patients predominantly underwent bortezomib-based chemotherapy. The ANDROMEDA trial is presently the largest randomized controlled trial comparing daratumumab-based therapy with bortezomib-based therapy in terms of prognosis improvement for AL amyloidosis patients, including 388 cases of AL amyloidosis patients from 109 centers [21]. This study revealed that patients treated with daratumumab-based therapy exhibited higher rates of hematologic CR, enhanced survival free from major organ deterioration, and reduced hematologic progression. In our subgroup analysis of patients with CA who did and did not receive daratumumab-based therapy, the OS rates in the daratumumab-based therapy group and the control group at the median follow-up were 85.7% (95% CI: 79.1–92.3) and 42.7% (95% CI: 35.5–49.9), respectively (see Appendix A). Therefore, daratumumab-based therapy can indeed improve outcomes in AL amyloidosis patients, including OS and hematologic CR.

### 4.4. Prognostic Value of Holter Monitoring in the Daratumumab Era

The extensive utilization of anti-CD38 monoclonal antibodies has led to favorable outcomes, including improved hematologic CR rates, more organ responses (particularly cardiac responses) in AL amyloidosis patients, and even enhanced one-year OS for stage IIIB patients [20,21]. Although further long-term follow-up is necessary, we expect a better OS beyond the first year of diagnosis with AL amyloidosis. Our study has demonstrated that AT and conduction delay in 24 h Holter monitoring are independent predictors of poor OS beyond the first year of diagnosis in patients with AL amyloidosis. Thus, 24 h Holter monitoring is a useful tool for risk stratification of patients with AL amyloidosis in the daratumumab era.

The application of ASCT is limited not only by stringent indications but also by the availability of resources and expertise among healthcare centers and patients’ socioeconomic conditions, especially in less-developed regions. Given that only ~20% of patients with AL amyloidosis may be able to undergo ASCT and the widespread use of anti-CD38 antibodies [41], we explored the prognostic value of Holter monitoring for OS in a cohort that mainly included non-transplant patients, among which 23.4% received daratumumab-based therapy and drew different conclusions than other studies. Thus, our findings are more generalizable to patients with AL amyloidosis than previous studies.

### 4.5. Prospects

Firstly, we identified two novel prognostic indicators for OS based on Holter findings, AT and conduction delay, which have not been previously reported. These findings suggest that Holter monitoring could serve as an effective and non-invasive tool for OS prediction in patients with AL. It provides incremental value beyond traditional prognostic models such as the Mayo 2012 staging system, enabling refined risk stratification. For AL patients with AT and/or conduction delay, closer follow-up and more aggressive treatment may be needed.

Secondly, the results of our study carry suggestive implications for future investigations. It is acknowledged that controversies exist regarding the optimal therapeutic strategies for AL patients with arrhythmias [35]. Our study reveals a poorer OS in AL patients with AT and/or conduction delay. However, due to limitations in sample sizes, we did not further investigate the potential benefits of pacemaker implantations and the use of anti-arrhythmic drugs to improve outcomes in these patients. Further in-depth research is needed to delineate the optimal treatment strategies for AL patients with AT and/or conduction delay.

### 4.6. Limitations

Our study was limited by its single-center, retrospective design. Prospective multicenter research studies might be necessary to confirm our findings and address potential biases introduced by the study’s design. Another limitation was potential selection bias. Specifically, 80.3% of study participants had cardiac involvement, indicating that clinicians may be more likely to order Holter monitoring for cardiac-involved patients. While it is important to note that a similar incidence of CA has been reported in the ANDROMEDA trial [21], the impact of this bias was relatively minimal. Lastly, the incidence of AF and NSVT in our study was relatively low, likely because we only included patients with newly diagnosed AL amyloidosis. Despite these limitations, our study presents innovative insights into the prognostic value of 24 h Holter monitoring for OS beyond the first year of diagnosis in a large cohort of patients with AL amyloidosis.

## 5. Conclusions

AT and conduction delay in continuous Holter monitoring are independent predictors of poor OS beyond the first year of diagnosis in patients with AL amyloidosis, providing additional information beyond staging with cardiac biomarkers and light chain burden. With the increased focus on beyond the first year of diagnosis in the anti-CD38 monoclonal antibody era, it is essential to perform Holter monitoring in patients with AL amyloidosis to identify and intervene with high-risk patients.

## Figures and Tables

**Figure 1 jcm-12-07457-f001:**
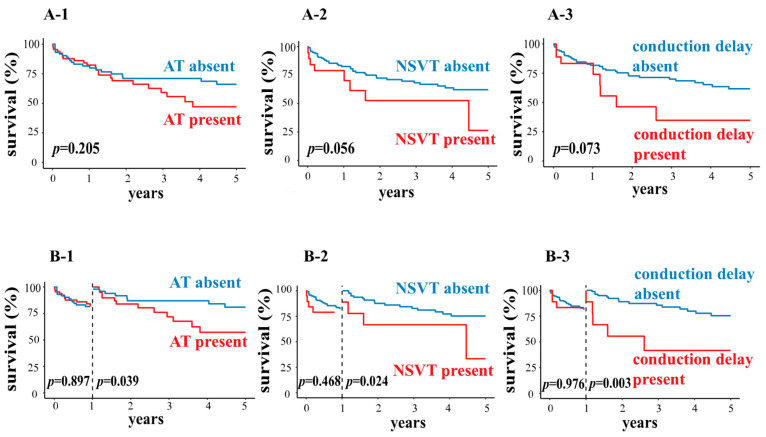
(**A1**–**A3**) Kaplan–Meier survival estimates for OS based on the presence or absence of AT, NSVT, and conduction delay. (**B1**–**B3**) Landmark analysis estimates for OS within the 0–1 year and 1–5 year time frames based on the presence or absence of AT, NSVT, and conduction delay. Note: AT: atrial tachycardia; NSVT: non-sustained ventricular tachycardia; OS: overall survival.

**Table 1 jcm-12-07457-t001:** Demographic and clinical characteristics (*n* = 137).

Variable	Value
Male—no. (%)	102 (74.5)
Age at diagnosis—year (mean ± SD)	61.9 (51.7–72.1)
**Comorbidities—no. (%)**	
Coronary artery disease	14 (10.2)
Hypertension	46 (33.6)
Diabetes	17 (12.4)
Chronic kidney disease	1 (0.7)
**Laboratory tests**	
λ Light chain restricted—no. (%)	103 (75.2)
dFLC—mg/dL (IQR)	190.5 (82.1–472.2)
NTproBNP—pg/mL (IQR)	1101 (329.3–4807.8)
Troponin I—ng/mL (IQR)	0.03 (0.01–0.09)
**Involved organs—no. (%)**	
Heart	110 (80.3)
Kidney	98 (71.5)
Liver	12 (8.8)
Others	46 (33.6)
Multi-organ *	101 (73.7)
**Mayo 2012 stages—no./total (%)**	
I	38/128 (27.7)
II	33/128 (24.1)
III	33/128 (24.1)
IV	24/128 (17.5)
**First-line treatment—no. (%)**	
ASCT based therapy	6 (4.4)
Non-transplant therapy	131 (95.6)
Daratumumab-based chemotherapy	32 (23.4)
Bortezomib-based chemotherapy	70 (51.1)
Other chemotherapy	21 (15.3)
**Echocardiographic findings**	
EF—% (IQR)	65 (58.9–71)
IVS—mm (IQR)	1.1 (0.9–1.4)

Note: ASCT: autologous hematopoietic stem cell transplantation; dFLC: difference in the free light chain; EF: ejection fraction; IQR: interquartile range; IVS: intraventricular septal thickness; NTproBNP: N terminal pro-brain natriuretic peptide; SD: standard deviation. *: ≥2 organs involved.

**Table 2 jcm-12-07457-t002:** Holter findings and anti-arrhythmic therapies (*n* = 137).

Variable	Value
**Holter findings**	
Average heart rate—beats/min (IQR)	80 (70–90)
Atrial couplets present—no. (%)	76 (55.5)
% of total beats (IQR)	0.01 (0–0.02)
AT present—no. (%)	65 (47.4)
AT runs—median (IQR)	1 (1–3)
Longest-beat AT run—median (IQR)	7 (3.8–11)
AF present—no. (%)	5 (3.6)
% of total beats (IQR)	100 (50.0–100)
AVRT/AVNRT present—no. (%)	0 (0)
VC present—no. (%)	30 (21.9)
% of total beats (IQR)	0.01 (0–0.05)
NSVT present—no. (%)	19 (13.9)
NSVT runs—median (IQR)	1 (1–3)
Longest-beat NSVT run—median (IQR)	5 (3–10)
Basic rhythm—no. (%)	
Sinus	116 (84.7)
Sinus with conduction delay	17 (12.4)
AF	4 (2.9)
VT/VF	0 (0)
Pacemaker	0 (0)
**Anti-arrhythmic therapies—no. (%)**	
Anti-arrhythmic drugs	23 (16.8)
Beta-blockers	21 (15.3)
Amiodarone	1 (0.7)
Ivabradine	1 (0.7)
Pacemaker implantation	4 (2.9)

Note: AF: atrial fibrillation; AT: atrial tachycardia; AVRT/AVNRT: atrial ventricular reciprocating tachycardia/atrial ventricular nodal reentrant tachycardia; IQR: interquartile range; NSVT: non-sustained ventricular tachycardia; VC: ventricular couplets; VF: ventricular fibrillation; VT: ventricular tachycardia.

**Table 3 jcm-12-07457-t003:** Holter findings and anti-arrhythmic therapies in CA/non-CA patients.

Variable	CA Patients (*n* = 110)	Non-CA Patients (*n* = 27)	*p*-Value
**Holter findings—no. (%)**			
Atrial couplets present	62 (56.4)	14 (51.9)	0.673
AT present	58 (52.7)	7 (25.9)	**0.012**
AF present	4 (3.6)	1 (3.7)	1.000
VC present	27 (24.5)	3 (11.1)	0.130
NSVT present	18 (16.4)	1 (3.7)	0.163
Basic rhythm—no. (%)			0.703 *
Sinus	92 (83.6)	24 (88.9)	
Sinus with con duction delay	15 (13.6)	2 (7.4)	
AF	3 (2.7)	1 (3.7)	
VT/VF	0 (0)	0 (0)	
Pacemaker	0 (0)	0 (0)	
**Anti-arrhythmic therapies—no. (%)**			0.245 ^$^
Anti-arrhythmic drugs	20 (18.2)	3 (11.1)	
Beta-blockers	19 (17.3)	3 (11.1)	
Amiodarone	1 (0.9)	0 (0)	
Ivabradine	1 (0.9)	0 (0)	
Pacemaker implantation	4 (3.6)	0 (0)	

Note: CA: cardiac amyloidosis; AF: atrial fibrillation; AT: atrial tachycardia; NSVT: non-sustained ventricular tachycardia; VC: ventricular couplets; VF: ventricular fibrillation; VT: ventricular tachycardia. *: *p*-value for sinus vs. non-sinus rhythm. ^$^: *p*-value for receive vs. not receive anti-arrhythmic therapies.

**Table 4 jcm-12-07457-t004:** Overall survival based on Holter findings (*n* = 137).

Variable	0–5 Years OS (95% CI)	*p*-Value	0–1 Year OS (95% CI)	*p*-Value	1–5 Years OS * (95% CI)	*p*-Value	Median OS (Months)	*p*-Value
**Atrial couplets**		0.395		0.156		0.768		0.656
Present	60.4 (53.3–67.5)		86.2 (82.1–90.3)		70.0 (62.5–77.5)		68.9	
Absent	57.0 (49.6–64.4)		76.5 (71.0–82.0)		74.4 (66.3–82.5)		NR **	
**AT**		0.205		0.897		**0.039**		0.107
Present	47.1 (38.4–55.8)		82.3 (77.4–87.2)		57.3 (47.2–67.4)		13.9	
Absent	66.1 (59.8–72.4)		81.5 (76.9–86.1)		81.0 (74.8–87.2)		NR **	
**AF**		0.828		0.184		0.330 ^$^		0.828
Present	60.0 (38.1–81.9)		60.0 (38.1–81.9)		100.0 (100.0–100.0) ^$^		NR **	
Absent	58.6 (53.3–63.9)		82.7 (79.3–86.1)		70.8 (65.1–76.5)		NR **	
**VC**		0.430		0.434		0.779		0.301
Present	62.0 (51.1–72.9)		80.0 (72.7–87.3)		77.5 (65.9–89.1)		39.2	
Absent	59.3 (53.7–64.9)		82.8 (79.1–86.5)		71.6 (65.6–77.6)		NR **	
**NSVT**		0.057		0.471		**0.024**		0.061
Present	26.3 (6.4–46.2)		78.9 (69.5–88.3)		33.3 (8.5–58.1)		22.4	
Absent	62.2 (56.9–67.5)		82.6 (79.0–86.2)		75.3 (69.8–80.8)		NR **	
**Conduction delay**		0.073		0.974		**0.003**		0.135
Present	34.7 (19.8–49.6)		83.3 (74.5–92.1)		41.7 (24.4–59.0)		12.1	
Absent	61.7 (56.3–67.1)		81.8 (78.2–85.4)		75.4 (69.7–81.1)		NR **	

Note: AF: atrial fibrillation; AT: atrial tachycardia; CI: confidence interval; NSVT: non-sustained ventricular tachycardia; OS: overall survival; VC: ventricular couplets; NR: not reached. *: Only patients still alive at 1 year were included because the landmark analysis was undertaken at 1 year. ^$^: None of the 3 patients with AF died in 1–5 years. **: The overall mortality rate did not reach 50% at the end of the follow-up.

**Table 5 jcm-12-07457-t005:** Multivariate survival analyses *.

Variable	Overall Survival—HR (95% CI)	*p*-Value
Age > 60 yr	1.5 (0.5–4.6)	0.483
**AT, present vs. absent**	2.6 (1.0–6.5)	**0.049**
Mayo 2012 Stage II vs. I	3.3 (1.0–10.9)	0.05
Mayo 2012 Stage III vs. I	1.0 (0.2–5.4)	0.978
Mayo 2012 Stage IV vs. I	6.7 (1.7–26.0)	0.006
Age > 60 yr	1.7 (0.6–5.2)	0.337
**NSVT, present vs. absent**	2.0 (0.6–7.1)	0.275
Mayo 2012 Stage II vs. I	3.7 (1.1–12.0)	0.033
Mayo 2012 Stage III vs. I	1.2 (0.2–6.6)	0.844
Mayo 2012 Stage IV vs. I	5.0 (1.2–21.5)	0.032
Age > 60 yr	1.9 (0.6–5.7)	0.273
**Conduction delay, present vs. absent**	4.3 (1.3–14.3)	**0.016**
Mayo 2012 Stage II vs. I	5.1 (1.5–18.1)	0.011
Mayo 2012 Stage III vs. I	1.7 (0.3–10.3)	0.541
Mayo 2012 Stage IV vs. I	7.8 (1.9–31.5)	0.004

Note: AT: atrial tachycardia; HR: hazard ratios; NSVT: non-sustained ventricular tachycardia; OS: overall survival; VC: ventricular couplets. *: Only patients still alive at 1 year were included because the landmark analysis was undertaken at 1 year.

## Data Availability

The data supporting this study’s findings are available from the corresponding authors, Jian Liu and Jin Lu, upon reasonable request.

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
