# Peer review of "Prognostic Value of Holter Monitoring in Light Chain Amyloidosis"

_jcm, 2023, doi:10.3390/jcm12237457_

Round 1

Reviewer 1 Report

Comments and Suggestions for Authors

Authors reported the prognostic value in survival using Holter monitoring in patients with light chain amyloidosis. The study is overall interesting, but some information must need be provided.

Major:

1, Does those patients with arrhythmias receive appropriate  anti-arrhythmic therapies? If so, which drug was used? Is there any difference between different groups? This information must be provided.

2, Why there is only 27 patients in the non-CA group, compared to the 110 in the CA group? Is there any difference of survival in different groups?

Minor:

1, The font size in Figure 1 is way too small.

2, More information should be provided regarding AL amyloidosis associated arrhythmias in the introduction.

3, More details regarding inclusion and exclusion criteria should be provided.

4, Writing skills need to be improved and details need to be included at important places. For example, in the abstract "To evaluate the prognostic value of Holter monitoring of patients with 18 light chain amyloidosis (AL amyloidosis)". What does authors want to express? The prognostic value in what? There are also many other similar inaccuracy in the manuscript.

5, What's the translational value of authors' finding? Will it guide the anti-arrhythmic therapy in those patients? This should be especially discussed.

Comments on the Quality of English Language

Writing must be improved.

Reviewer 2 Report

Comments and Suggestions for Authors

The authors present a study showing the relationship between the presence of arrhythmias on Holter monitoring and overall survival in patients with AL amyloidosis. This is a well-designed study for which the conclusions support the results and analysis. 

There is some redundancy in the sub-analysis of those with cardiac involvement and Mayo staging. Given 80% had cardiac amyloid, showing the analysis with early and advanced stage Mayo criteria are redundant. The survival analysis is displayed for the later (Table 3) but not for the former subgrouping. 

Why do the authors believe the prevalence of atrial fibrillation was so low?

There is an implication that as 20% received anti-CD38 therapy, it would account for the discrepancy in overall survival in the first year after diagnosis. Can the authors provide a subgroup analysis of patients with CA that did and did not receive anti-CD38 therapy to bolster the former conclusion (sample size may be small)? This would provide merit as to why this finding conflicted with Palladini et al. 

Reviewer 3 Report

Comments and Suggestions for Authors

Dear Authors,

I would like to point out that the article is well written and I enjoyed reading it. However, it would contribute to the article if the authors clarified some sections. These;

1. Introduction: The introduction section should be expanded by adding information about the disease states in which AL amyloidosis is seen, its clinical course, diagnostic procedures, laboratory and treatments.

2. Method: When choosing the patient group, the authors determined the population size, margin of error, and the power analysis of the patient group they would include in the study. It needs to be explained.

3. What were the comorbid conditions of the patients?

4. What were the patients' acceptance and rejection criteria?

5. Statistics: Which of the One-sample Kolmogorov Smirnov test or Shapiro-Wilk Test was used when the authors determined the significance status?

Round 2

Reviewer 1 Report

Comments and Suggestions for Authors

Authors answered all my questions. I don't have further concerns.

Comments on the Quality of English Language

Now the quality is much better.

Reviewer 2 Report

Comments and Suggestions for Authors

The authors' responses are excellent. All concerns have been addressed. I am especially content that the daratumumab survival analysis was completed and included here as it significantly bolsters the findings. 

Reviewer 3 Report

Comments and Suggestions for Authors

Dear Authors,

It can be seen that the authors made the desired changes. I want to thank the authors for their good-faith work. I think the article is more elite in this form.

Kind regards.